# Shifting Precipitation Patterns Drive Growth Variability and Drought Resilience of European Atlas Cedar Plantations

Jesús Julio Camarero [1,*], Antonio Gazol [1], Michele Colangelo [1,2], Juan Carlos Linares [3], Rafael M. Navarro-Cerrillo [4], Álvaro Rubio-Cuadrado [5], Fernando Silla [6], Pierre-Jean Dumas [7] and François Courbet [7]

1. Instituto Pirenaico de Ecología (IPE-CSIC), 50192 Zaragoza, Spain; agazolbu@gmail.com (A.G.); michelecolangelo3@gmail.com (M.C.)
2. School of Agricultural, Forest, Food and Environmental Sciences (SAFE), University of Basilicata, 85100 Potenza, Italy
3. Departamento de Sistemas Físicos, Químicos y Naturales, Universidad Pablo de Olavide, 41013 Sevilla, Spain; jclincal@upo.es
4. Laboratorio de Selvicultura, Departamento Ingeniería Forestal, Dendrocronología y Cambio Climático, DendrodatLab-ERSAF, Campus de Rabanales, Universidad de Córdoba, Crta. IV, km. 396, 14071 Cordova, Spain; rmnavarro@uco.es
5. Departamento de Sistemas y Recursos Naturales, Escuela Técnica Superior de Ingeniería de Montes, Forestal y del Medio Natural, Universidad Politécnica de Madrid, Ciudad Universitaria s/n, 28040 Madrid, Spain; alvaro.rubio.cuadrado@upm.es
6. Departamento Biología Animal, Parasitología, Ecología, Edafología y Química Agrícola, University Salamanca, 37007 Salamanca, Spain; fsilla@usal.es
7. INRAE, UR 629 Ecologie des Forêts Méditerranéennes, URFM, 84914 Avignon, France; pierre-jean.dumas@inrae.fr (P.-J.D.); francois.courbet@inrae.fr (F.C.)
* Correspondence: jjcamarero@ipe.csic.es; Tel.: +34-976-363-222 (ext. 880041)

**Abstract:** Tree plantations have been proposed as suitable carbon sinks to mitigate climate change. Drought may reduce their carbon uptake, increasing their vulnerability to stress and affecting their growth recovery and resilience. We investigated the recent growth rates and responses to the climate and drought in eight Atlas cedar (*Cedrus atlantica*) plantations located along a wide climate gradient from wetter sites in south-eastern France and north Spain to dry sites in south-eastern Spain. The cedar growth increased in response to the elevated precipitation from the prior winter to the current summer, but the influence of winter precipitation on growth gained importance in the driest sites. The growth responsiveness to climate and drought peaked in those dry sites, but the growth resilience did not show a similar gradient. The Atlas cedar growth was driven by the total precipitation during the hydrological year and this association strengthened from the 1980s onwards, a pattern related to the winter North Atlantic Oscillation (NAO). High winter NAO indices and drier conditions were associated with lower growth. At the individual level, growth resilience was related to tree age, while growth recovery and year-to-year growth variability covaried. Plantations' resilience to drought depends on both climate and tree-level features.

**Keywords:** *Cedrus atlantica*; dendroecology; drought; growth resilience; plantations





## 1. Introduction

Climate models forecast harsher conditions for many conifer forests due to a higher recurrence of extreme climatic events such as droughts, which will negatively impact their productivity, reducing tree growth and constraining growth resilience [1,2]. If droughts become more severe and frequent, recurrent stressing conditions may also reduce their long-term growth recovery [3], worsening forest health and rising mortality rates [4,5]. A reduced post-drought growth recovery leads to a declining resilience due to drought legacies or carryover effects, which challenge the ability of forests to act as effective carbon sinks, limiting their potential to mitigate climate warming [6,7].

Planting trees through massive afforestation programs has been proposed to increase the potential of forests to mitigate climate change [8,9]. The use of planted conifers with high growth resilience could be an adaptive strategy to cope with the regional dryness trends in drought-prone areas, albeit restoring or protecting natural forests would allow more lasting carbon storage [10,11]. However, severe dry spells could surpass the threshold of some drought-tolerant conifers, reducing the productivity at a regional scale [12,13], particularly where long-term warming is expected to occur concomitant to more intense and frequent droughts, such as in south-western Europe [14]. Hence, reliable field assessments of growth resilience across wide climatic gradients are needed to quantify the role played by conifer plantations in climate change mitigation.

In Europe, cedar species have been considered drought-resilient species being planted since the late 19th century [15,16]. For instance, the Atlas cedar (*Cedrus atlantica* Manetti ex Carrière) shows high growth resilience in its natural habitats in north-western Africa, where it copes with climatic conditions forecasted for continental, drought-prone areas of south-western Europe during the late 21st century [17,18]. Nevertheless, recent severe droughts have also reduced the productivity of these forests, while dieback episodes have been observed in the xeric Atlas cedar forests of Morocco and Algeria [19–23]. Therefore, the Atlas cedar plantations performed in south-western Europe might also be sensitive to coming drought-induced stress. Therefore, evaluating the growth resilience in Atlas cedar plantations may help to understand the sensibility and vulnerability of mountain coniferous plantations to future hotter droughts. In addition, this would aid to establish which planted stands could be more or less vulnerable under a warmer and drought-recurrent scenario.

Individual variables such as tree size and age or local site conditions may override the competition as drivers of growth resilience [24]. Here, we studied eight Atlas cedar plantations encompassing wide geographic and climatic gradients to assess the relative contributions of the local climate and tree features to growth resilience. Plantations provide a suitable setting to investigate among-sites differences and tree-level responses to environmental conditions. We aimed: (i) to reconstruct the radial growth of European Atlas cedar plantations, (ii) to quantify how climate, and particularly drought severity, impacts growth and its resilience capacity, and (iii) to investigate the roles played by individual features such as the tree age, growth rate and year-to-year growth variability in resilience. We hypothesize that those plantations situated in the driest sites display the highest sensitivity to water shortage, but also the highest resilience (the capacity to recover growth values similar to those observed before the drought, cf. [25]) and the lowest resistance (i.e., the amount of growth reduction due to drought). At the individual level, we expect that those trees showing a higher growth rate and age, and also higher growth variability, a surrogate of individual sensitivity to climate variability, will show the highest resilience. Alternatively, older or larger trees, with less efficient hydraulic or photosynthetic systems than younger or smaller trees [18,26], would present a lower resilience and be slower to reach pre-drought growth levels.

## 2. Materials and Methods

### 2.1. Study Sites: Atlas Cedar Plantations in South-Western Europe

We studied eight Atlas cedar (*C. atlantica*) plantations distributed from south-western France (Valliguières) to south-eastern Spain (Puerto de la Mora) (Table 1, Figure S1 in Supporting Information). They are located along wide geographical (latitude range 37.3°–44.0° N, longitude range 5.7° W–4.6° E), altitudinal (256–1780 m a.s.l.) and climatic gradients (mean annual temperature 10.5 °C–14.4 °C, annual precipitation 390–829 mm, annual water balance −620 to 156 mm) (Table 1, Figure S2). Most sites are located on acid soils of the Cambisol type developed on schist and shale, excepting the Sariñena and Valliguières sites where soils are basic, developed on limestone. Besides, the soils in Sariñena may contain gypsum.

**Table 1.** Features of the Atlas cedar plantations studied in France and Spain. See site's climate diagrams in Figure S1.

| Site (Code) | Latitude N | Longitude (−W, +E) | Elevation (m a.s.l.) | Mean Annual Temperature (Mean Minimum–Maximum Temperatures) (°C) | Annual Precipitation (mm) | Annual Water Balance (mm) |
|---|---|---|---|---|---|---|
| Valliguières (MV) | 44.019 | 4.624 | 256 | 14.0 (−1.8–26.5) | 699 | −66 |
| Berriozar (CN) | 42.844 | −1.664 | 450 | 12.6 (−0.1–24.7) | 829 | 156 |
| Sariñena (SA) | 41.783 | −0.222 | 295 | 14.4 (−0.5–30.5) | 390 | −620 |
| Bañón (CB) | 40.800 | −1.179 | 1358 | 13.0 (−1.0–26.5) | 423 | −386 |
| Monte Abantos (AC) | 40.609 | −4.146 | 1325 | 12.0 (−0.1–28.0) | 526 | −158 |
| Monte Mario-Béjar (CE) | 40.384 | −5.749 | 1010 | 11.0 (0.8–29.5) | 507 | −144 |
| Baza (BA) | 37.305 | −2.908 | 1780 | 10.5 (0.1–26.4) | 470 | −295 |
| Puerto de la Mora (PM) | 37.286 | −3.458 | 1380 | 13.8 (0.5–28.6) | 411 | −470 |

In north-western Africa, Atlas cedars are exposed to annual mean temperatures ranging from 7.5 °C to 15 °C (mean minimum temperatures for the coldest month between −1 °C and −8 °C and mean maximum temperatures for the warmest month from 25 °C to 35 °C) [27]. In their native African habitat, summer droughts last from 2 to 4 months and soils are mainly developed on limestone [17]. In south-western Europe, Atlas cedar plantations show an optimum growth in wet sites (annual precipitation from 800 up to 1500 mm) and on relatively deep (45–120 cm) soils [15]. According to [15], the Atlas cedar does not grow well on loamy and sandy soils, unless the soil is deep enough. The Atlas cedar is sensitive to a water deficit due to moderate stomatal conductance rates, which are compensated by updating deep water resources through their efficient root systems [28,29]. The selected plantations were not affected by severe thinning treatments since the 1980s. Therefore, we assumed that the tree-to-tree competition remained stable over the period of 1980–2020.

*2.2. Climate Data, Climate and Atmospheric Circulation Indices, and Drought Index*

We used monthly and seasonal mean maximum and minimum temperature and summed precipitation data (period 1950–2020) from the 1 km$^2$ gridded E-OBS v. 22.0e database [30]. To obtain a global assessment of precipitation–growth associations, we also obtained a regional hydrological-year precipitation for the region delimited by the coordinates 5.75° W–1.00° E and 37.00°–44.25° N by summing monthly precipitation data from the prior October up to the current September. In the period 1980–2020, the mean precipitation over this region was 455 mm and the years with the lowest precipitation were 1995 (333 mm), 2005 (283 mm) and 2012 (337 mm).

We calculated the annual climatic water balance as the difference between the precipitation and potential evapotranspiration and the actual evapotranspiration because these variables are useful proxies of tree growth rates and productivity, respectively [31]. The potential evapotranspiration was calculated using a modified Thornthwaite method following [32]. This variable was estimated using the software AET-calculator developed by D.G. Gavin and available at https://pages.uoregon.edu/dgavin/software.html, accessed 4 October 2021 (see Figure S2).

Several climate indices accounting for the major atmospheric circulation patterns over the study area were considered. Specifically, the monthly and seasonal data of the North Atlantic Oscillation (NAO) index, the Southern Oscillation Index (SOI) and the Western Mediterranean Oscillation (WeMO) index were obtained. The NAO is negatively correlated

with winter precipitation over the western Mediterranean Basin [33]. The SOI reflects the influence of the El Niño–Southern Oscillation on eastern and south-eastern Iberia with positive correlations between the spring to summer precipitation and SOI values [34]. The WeMO accounts for autumn and winter precipitation patterns over north- eastern Iberia with positive WeMO values phases corresponding to low precipitation [35]. The NAO and SOI data were downloaded from the Climate Explorer website (https://climexp.knmi.nl, accessed 4 October 2021), and the WeMO index was obtained from the Climatology Group website of the University of Barcelona (http://www.ub.edu/gc/wemo/, accessed 4 October 2021).

The Standardized Precipitation and Evapotranspiration Index (SPEI) was used to characterize the drought severity at different temporal resolutions with high and low values corresponding to wet and dry conditions, respectively [36]. The gridded (0.5° resolution) SPEI values calculated at time scales from 1 to 24 months were obtained for each site from the global SPEI database available at https://spei.csic.es/database.html (accessed 4 October 2021).

### 2.3. Field Sampling and Tree-Ring Width Data

In each site, we randomly selected and sampled from 12 to 33 trees (Table 2). Their diameter at breast height (Dbh) was measured using tapes. No recent dieback symptoms or outbreak impacts (e.g., severe defoliation) were observed in the sampled trees. Two opposite cores were taken at 1.3 m from each tree, always perpendicular to the maximum slope, using Pressler increment borers. The cores were air dried, glued onto wooden supports and carefully sanded (using sandpapers from 300 to 600 grit) to distinguish the annual rings. Then, the cores were visually cross-dated by listing and matching very narrow rings, usually associated to droughts, among a series of coexisting trees [37].

The tree-ring width was measured to 0.01 mm precision in the dated cores using a LINTAB device and TSAP-Win software (Rinntech, Germany) in all sites excepting the Valliguières site. In this site, ring widths were measured using the WinDENDRO™ system (Regent Instruments Inc., Ste.-Foy, QC, Canada). The cross-dating quality was further checked using the COFECHA software, which calculates the mean correlations between individual series and a mean site series [38].

The tree-ring width data were transformed into a basal area increment (BAI), which is biologically more meaningful to quantify growth trends as changes in a conductive xylem area [39]. Annual BAI values were calculated using tree-ring width data as the difference between consecutive annual cross-sectional basal areas assuming a circular shape of stems. The mean BAI of the two cores for each tree was computed. Finally, the tree age at 1.3 m was estimated as the number of rings for cores with pith or presenting curved innermost rings located close to the pith.

**Table 2.** Tree size (Dbh, diameter at breast height) and radial-growth values and dendrochronological statistics obtained for the sampled Atlas cedar plantations. Values are means ± SE. The last column shows the percentage of growth variability ($R^2$prec) explained by the precipitation of the hydrological year (prior October to current September) based on a linear regression with the mean site residual chronologies considering the common period 1981–2017. In all cases, the regression coefficients were significant ($p < 0.05$).

| Site | Dbh (cm) | Age at 1.3 m (Years) | No. Trees (No. Radii) | Tree-Ring Width (mm) | First-Order Autocorrelation (AC1) | Mean Sensitivity (MSx) | Correlation with Master | Best Replicated Period | Frequency of Negative Pointer Years (%) | $R^2$prec (%) |
|---|---|---|---|---|---|---|---|---|---|---|
| MV | 32.1 ± 1.5 | 39 ± 3 | 8 (16) | 2.30 ± 0.07 | 0.79 ± 0.01 | 0.28 ± 0.01 | 0.69 ± 0.01 | 1977–2017 | 6.8 ± 1.9 | 27.3 |
| CN | 39.0 ± 2.1 | 69 ± 2 | 12 (24) | 2.20 ± 0.08 | 0.73 ± 0.02 | 0.37 ± 0.01 | 0.81 ± 0.02 | 1957–2020 | 17.0 ± 3.8 | 32.4 |
| SA | 36.8 ± 3.0 | 36 ± 1 | 12 (24) | 4.00 ± 0.25 | 0.69 ± 0.04 | 0.26 ± 0.02 | 0.60 ± 0.02 | 1981–2020 | 11.5 ± 1.8 | 15.0 |
| CB | 28.3 ± 0.9 | 53 ± 1 | 19 (38) | 2.22 ± 0.06 | 0.59 ± 0.02 | 0.49 ± 0.01 | 0.88 ± 0.01 | 1967–2020 | 21.0 ± 4.9 | 45.1 |
| AC | 43.4 ± 2.3 | 65 ± 2 | 17 (34) | 2.53 ± 0.15 | 0.65 ± 0.09 | 0.31 ± 0.01 | 0.62 ± 0.02 | 1957–2020 | 13.6 ± 2.2 | 22.5 |
| CE | 41.6 ± 1.0 | 58 ± 1 | 16 (32) | 2.91 ± 0.09 | 0.65 ± 0.03 | 0.41 ± 0.01 | 0.80 ± 0.01 | 1965–2020 | 15.0 ± 3.3 | 47.7 |
| BA | 26.3 ± 1.4 | 36 ± 1 | 16 (32) | 2.94 ± 0.16 | 0.54 ± 0.04 | 0.34 ± 0.02 | 0.75 ± 0.02 | 1981–2020 | 12.2 ± 2.6 | 61.3 |
| PM | 35.5 ± 1.1 | 45 ± 1 | 15 (30) | 2.69 ± 0.06 | 0.81 ± 0.01 | 0.32 ± 0.01 | 0.80 ± 0.02 | 1974–2020 | 19.1 ± 3.6 | 38.6 |

### 2.4. Processing Tree-Ring Data: Ring-Width Indices and Climate–Growth Relationships

To calculate climate–growth relationships, ring-width indices were calculated. The tree-ring width data were detrended and standardized to account for changes in tree size, age and stand dynamics affecting growth [40]. The detrending was done by fitting a 67% cubic smoothing spline with a 50% cut-off frequency, which allowed removing long- to mid-term growth variability [40]. Then, standard, dimensionless ring-width indices (RWI) were obtained by dividing observed by fitted ring-width data. It should be noted that an autoregressive modeling was not performed on the resulting detrended series of RWI to keep the year-to-year persistence in growth. Lastly, the individual ring-width indices were averaged into mean sites series (chronology) using a bi-weight robust mean [40]. These procedures were done using the ARSTAN software ver. 4.44 [41].

Several tree-ring statistics were calculated at the tree level to characterize individual growth variability [42]. First, the mean and first-order autocorrelation (AC1), a measure of growth persistence, of ring-width data were calculated. Second, the mean sensitivity of RWIs (MSx), which reflects the relative change in width among consecutive rings [40], was obtained. Third, we calculated the mean correlation (Corr) of individual RWI series with the site mean series, which accounts for within-site growth synchrony.

The climate-growth relationships were assessed by calculating Pearson correlations between monthly or seasonal climate variables (mean maximum and minimum temperatures, total precipitation) or indices (NAO, SOI and WeMO) and the sites' mean series of ring-width indices, which followed normal distributions. The correlations were calculated from prior to current September and the 0.05 and 0.01 significance levels were considered. Analogously, Pearson correlations were calculated between the monthly SPEI values calculated at time scales from 1 to 24 months and the sites' ring-width series. We obtained the percentage of growth variability ($R^2$prec) explained by the precipitation of the hydrological year in each site.

To assess the temporal changes in precipitation–growth relationships for the whole study region, the hydrological-year precipitation for the region delimited by the coordinates 5.75° W–1.00° E and 37.00°–44.25° N and the global mean of all sites' series were correlated. Moving Pearson correlations were calculated between the two variables considering 20-year moving intervals from 1961 to 2020. Furthermore, we used linear models to study the percentage of growth variability explained by precipitation.

Lastly, the relationships among sites' ring-width series were compared using Principal Component Analysis (PCA) calculated on the matrix of the standard chronologies. The resulting scores of the first (PC1) and second (PC2) principal components were plotted. Analogously, a second PCA was calculated on the matrix of Pearson correlations obtained by relating climate variables and site chronologies. This second PCA allowed characterizing the similarity of climate–growth relationships between sites. We also calculated climate–growth associations considering the first three PCs of the PCA calculated on the site chronologies. A similar analysis was carried out considering the three climate indices (NAO, SOI and WeMO).

### 2.5. Negative Pointer Years and Resilience Indices

We calculated the negative pointer years using the relative growth change method [43], which is based on the comparison of the growth (standard RWIs) in year *i* with the growth in the preceding *n* years. For every series, the negative pointer years were defined as those years with a growth lower than 40% of the average growth in the preceding three years.

We calculated the resilience components for every series across sites following [25]. These authors proposed three resilience components (i.e., resistance, recovery and resilience) to analyze the growth of individual trees prior, during and after extreme events. Resistance (Rt) is the difference between the growth in the drought year and that in the preceding n years. Recovery (Rc) measures the capacity of trees to recover growth after

drought. Resilience (Rs) quantifies the capacity to recover pre-drought growth levels. The indices were computed as:

$$Rt = RWI_D/RWI_{Pre-D} \tag{1}$$

$$Rc = RWI_{Post-D}/RWI_D \tag{2}$$

$$Rs = RWI_{Post-D}/RWI_{Pre-D} \tag{3}$$

where $RWI_D$ corresponds to the standard RWI during the drought year, and the $RWI_{Pre-D}$ and $RWI_{Post-D}$ correspond to the average RWI of the three years before and after the drought, respectively. The resilience components were calculated for every tree in each site in every year along the entire series length. The pointRes package [44] was used to calculate the pointer years and resilience components.

The interdependence existing between some of the resilience indices proposed by [25], as already noted by several studies [2,45–47] who showed the negative relationships between resistance and recovery, must be noted.

To characterize the relationships at the individual level among tree variables (Dbh, age), growth variables (mean tree-ring width, AC1, MSx, Corr, frequency of negative pointer years) and the resilience indices (Rt, Rc, Rs), Spearman non-parametric correlations were calculated.

### 2.6. Linear Mixed-Effects Models of Growth Resilience

We used linear mixed-effect models (LME) [48] to test for the relationship between resilience components (Rt, Rc, Rs) and tree growth features at the individual tree level. We ran separate models for each resilience component using the site as a random factor. The resilience components were modeled as a function of the tree age, mean tree-ring width for the entire series length and mean sensitivity. A full model containing the three variables as fixed effects was constructed and evaluated graphically. Recovery (Rc) was log-transformed ($\log(x+1)$) prior to the analyses to achieve normality assumptions.

We applied a multi-model inference approach to select the most informative models [49]. We ranked the models created with the different combinations of the three fixed effects according to their Akaike Information Criterion (AIC) values and selected the most parsimonious model among those models with a difference lower than 2 AIC units from the model with the lowest AIC. The fit of the final selected model was quantified by calculating the marginal ($R^2m$) and the conditional ($R^2c$) $R^2$ values which account for the variance explained by the fixed effects and by the fixed and random effects, respectively [50]. The LMEs were fitted using the nlme package [48], while the model selection was done with the MuMIn package [51]. All analyses were performed using the R software [52].

## 3. Results

### 3.1. Growth Patterns and Negative Pointer Years at Site Level

The mean Dbh within each site (indicated between parentheses) ranged between 26.3 (Baza) and 43.4 cm (Monte Abantos) and the mean age ranged between 36 (Sariñena, Baza) and 69 years (Berriozar) (Table 2). Accordingly, the mean tree-ring width varied from 2.2 (Berriozar) to 4.0 mm (Sariñena). The global averages (mean ± SE) of Dbh, age and tree-ring width were 35.3 ± 2.2 cm, 51 ± 4 years and 2.72 ± 0.21 mm, respectively. The maximum ring-width autocorrelation (AC1) values were observed at opposite extremes of the climatic gradient, i.e., in wet (Valliguières, AC1 = 0.79) and dry (Puerto de la Mora, AC1 = 0.81) sites. The highest mean sensitivity (MSx), mean correlation (Corr) and frequency of negative pointer years were found in the Bañón site, suggesting a high climatic stress. High Corr values were also found in the Berriozar, Monte Mario-Béjar and Puerto de la Mora sites.

The percentage of growth variability ($R^2prec$) explained by the precipitation of the hydrological year peaked in Baza (61.3%), followed by the Monte Mario-Béjar (47.7%) and Bañón (45.1%) sites. In contrast, wet sites located in the northernmost regions of the study area, such as Valliguières, Berriozar or Monte Abantos, presented low MSx and Corr values, a low frequency of negative pointer years and also a low R2prec (on average 27.4%).

Unexpectedly, the Sariñena site situated in a site under semi-arid climate conditions in north-eastern Spain showed low MSx and Corr values, few negative pointer years and the lowest $R^2$prec (15.0%).

The dependency of radial growth on tree size and age was clearly observed when considering BAI, which plateaued at a mean value of 16.0 cm$^2$ for an age of 35 years (Figure S3). The BAI (Figure S3) and RWI (Figure 1) data evidenced the drop in Atlas cedar growth during the 1995, 2005 and 2012 droughts in most sites.

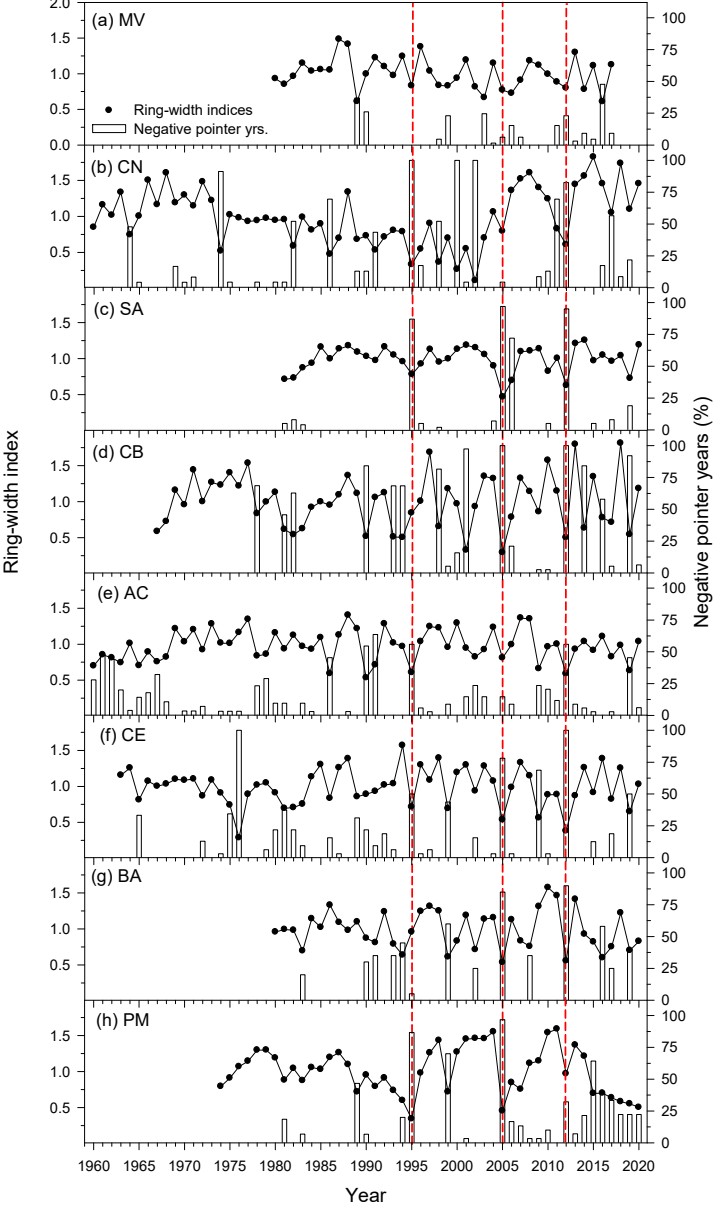

**Figure 1.** Site chronologies (mean series of ring-width indices) and percentage of negative pointer years (right y-axes) obtained for the studied Atlas cedar plantations: (**a**), MV, Valliguières; (**b**), CN, Berriozar; (**c**), SA, Sariñena; (**d**) CB, Bañón; (**e**) AC, Monte Abantos; (**f**) CE, Monte Mario-Béjar; (**g**) BA, Baza; and (**h**) PM, Puerto de la Mora. The dashed vertical lines indicate the 1995, 2005 and 2012 droughts.

### 3.2. Links of Growth with Climate Variability and Drought Severity at Site Level

High precipitation in the prior winter (e.g., Baza, Puerto de la Mora and Monte Mario-Béjar southern sites) and the current spring and summer (e.g., Valliguières, Berriozar, Monte Abantos and Bañón northern sites) enhanced Atlas cedar growth (Figure 2). The

most influential month for growth during the growing season and regarding precipitation shifted from May (e.g., Puerto de la Mora, Monte Abantos) to June (Berriozar) and even August (Valliguières) northwards. Warm conditions from May to July were related to decreased growth, whilst high minimum temperatures increased growth, particularly in the southernmost and driest sites. High February and April minimum temperatures were also associated with increased growth in Sariñena and Berriozar, respectively. These two sites were located in northern Spain (Table 1).

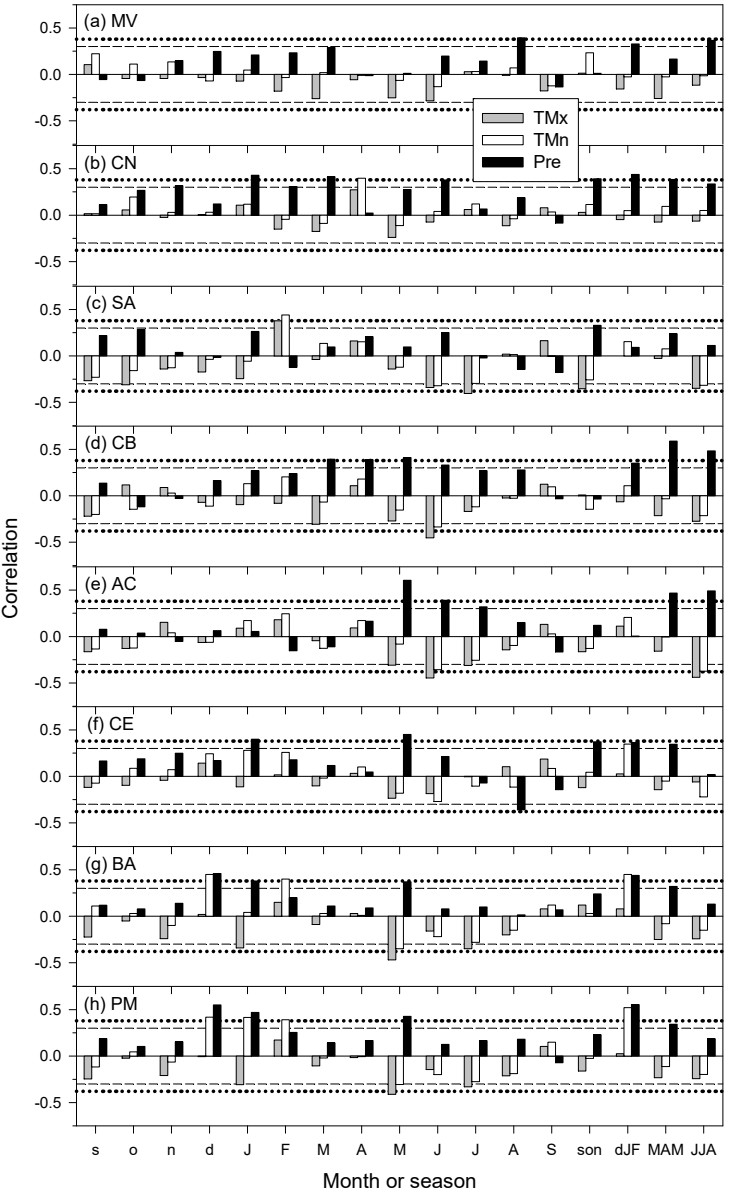

**Figure 2.** Climate–growth relationships assessed in eight Atlas cedar plantations ((**a**), MV, Valliguières; (**b**), CN, Berriozar; (**c**), SA, Sariñena; (**d**) CB, Bañón; (**e**) AC, Monte Abantos; (**f**) CE, Monte Mario-Béjar; (**g**) BA, Baza; and (**h**) PM, Puerto de la Mora). Bars show the Pearson correlations calculated by relating monthly or seasonal climate data (mean maximum—TMx—and minimum—TMn— temperatures, total precipitation—Pre) and sites' mean series of ring-width indices. Correlations were calculated from prior to current September and months abbreviated by lowercase and uppercase letters correspond to the previous and current years, respectively. Horizontal dashed and dotted lines indicate the 0.05 and 0.01 significance levels, respectively.

The PCA calculated on the site mean series of RWIs (Figure 3a) allowed separating along its first axis dry sites with high growth responsiveness to precipitation (e.g., Bañón, Baza Puerto de la Mora) from mesic or less responsive sites (e.g., Valliguières, Sariñena). The Berriozar site showed a component of local variability which was not observed in the other sites. The PCA calculated on climate–growth correlations (Figure 3b) discriminated southern, dry, responsive sites (e.g., Puerto de la Mora, Baza) from northern, less responsive, mesic sites (e.g., Valliguières, Berriozar).

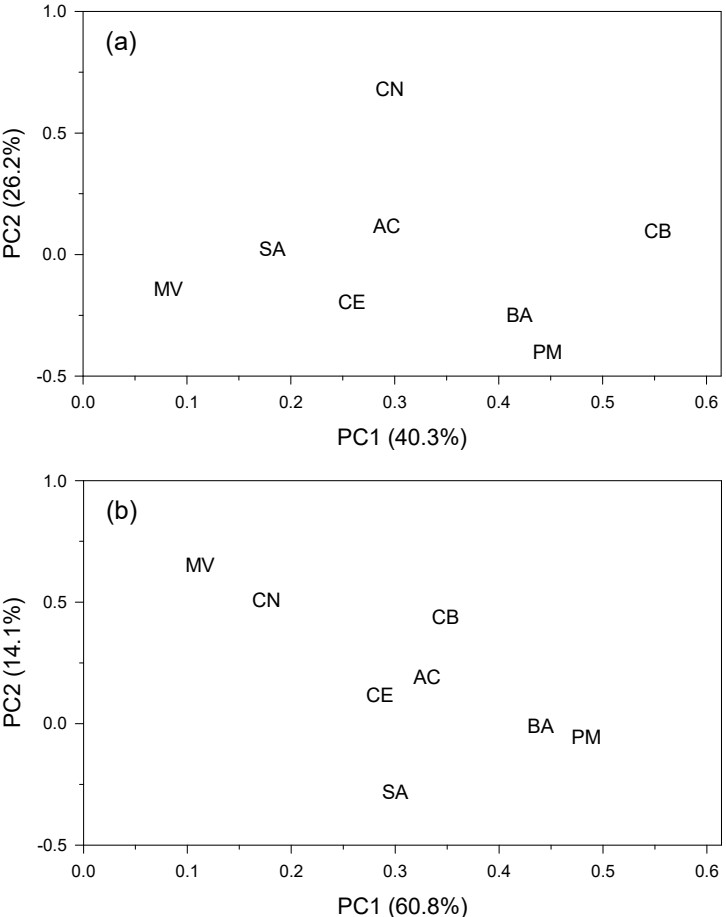

**Figure 3.** Biplots showing the scores in the first (PC1) and second (PC2) principal components of principal component analyses calculated on the matrix of (**a**) site chronologies (common period 1982–2011) and (**b**) climate–growth Pearson correlations of Atlas cedar plantations. Sites' codes: MV, Valliguières; CN, Berriozar; SA, Sariñena; CB, Bañón; AC, Monte Abantos; CE, Monte Mario-Béjar; BA, Baza; and PM, Puerto de la Mora. The plot (**a**) shows the similarity among sites based on site series of ring-width indices, whereas the plot (**b**) shows the similarity among sites based on climate–growth relationships. The percentages of variance explained by PC1 and PC2 are indicated. Climate–growth correlations were obtained by relating the site chronologies with mean monthly maximum and minimum temperatures and total precipitation from prior to current September.

The correlations with the SPEI drought index again revealed high Atlas cedar growth responsiveness to drought in the driest sites from south-eastern Spain, particularly in Puerto de la Mora and Baza (Figure 4). In these two sites, growth responded to long droughts (lasting from 4 to 24 months) throughout the year, i.e., corresponding to dry conditions in the winter previous to tree-ring formation. The Bañón site was also responsive to winter-spring droughts lasting from 4 to 8 months and occurring in June and July. Short (2–6 months SPEI) and long (4–18 months SPEI) droughts from March to September affected growth in Monte Abantos or Berriozar and Monte Mario-Béjar or Sariñena, respectively.

The northernmost Valliguières site showed the lowest growth–SPEI correlations that mainly occurred during late summer or early autumn months (Figure 4).

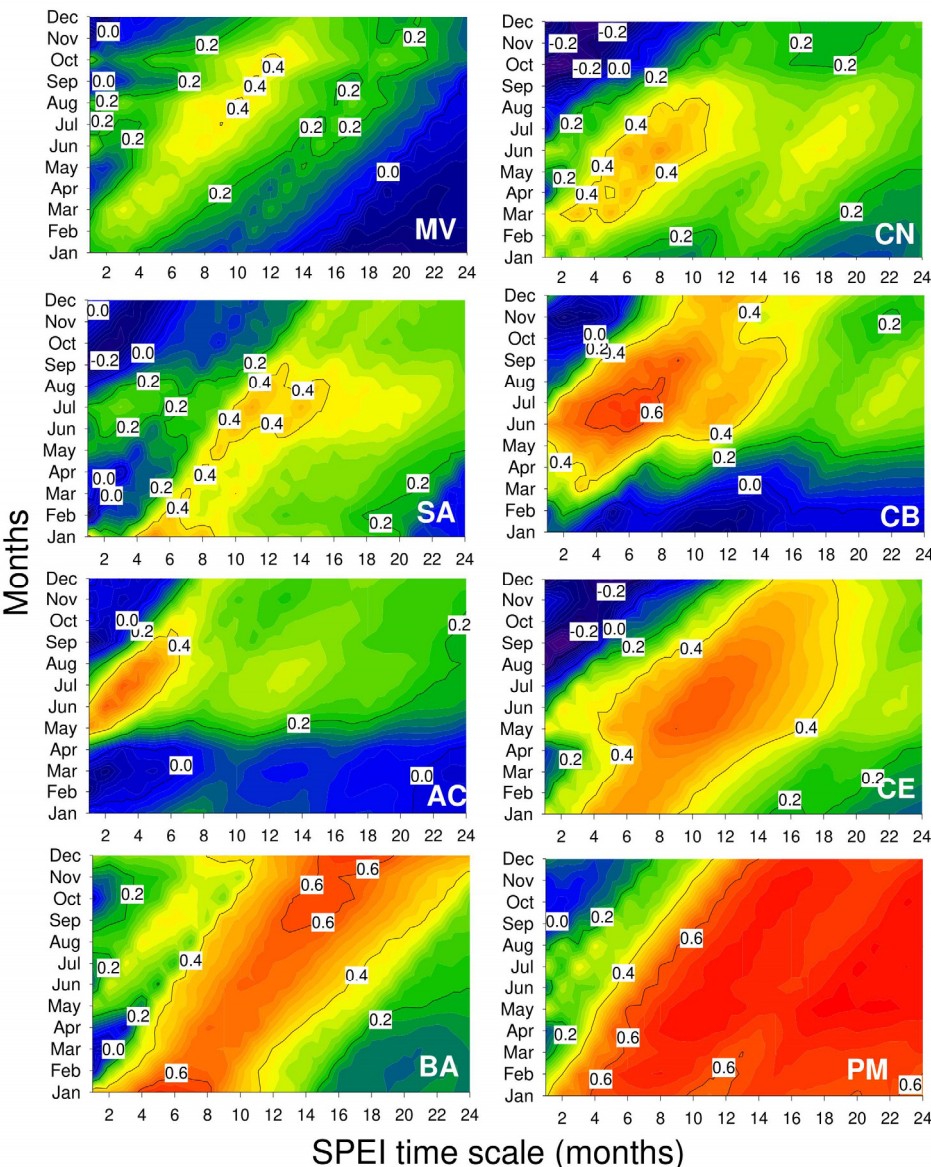

**Figure 4.** Correlations calculated between the Atlas cedar ring-width series and their corresponding monthly SPEI values (y-axes) at time scales from 1 to 24 months (x-axes). Significance levels are similar to those shown in Figure 2.

The Atlas cedar growth variability was tightly coupled with the regional precipitation of the hydrological year, from previous October to current September (Figure 5a). This precipitation was negatively related to the NAO of the same period ($r = -0.44$, $p = 0.004$) and also to the winter (December to February) NAO ($r = -0.53$, $p = 0.0003$). Atlas cedar growth was also negatively related to the winter NAO ($r = -0.31$, $p = 0.05$). Moreover, the moving correlation between precipitation and growth rate has increased through time ($\tau = 0.61$, $p < 0.001$), particularly after the 1995 drought. However, precipitation in the study area has not shown any significant trend (Figure 5a, $\tau = 0.06$, $p = 0.56$).

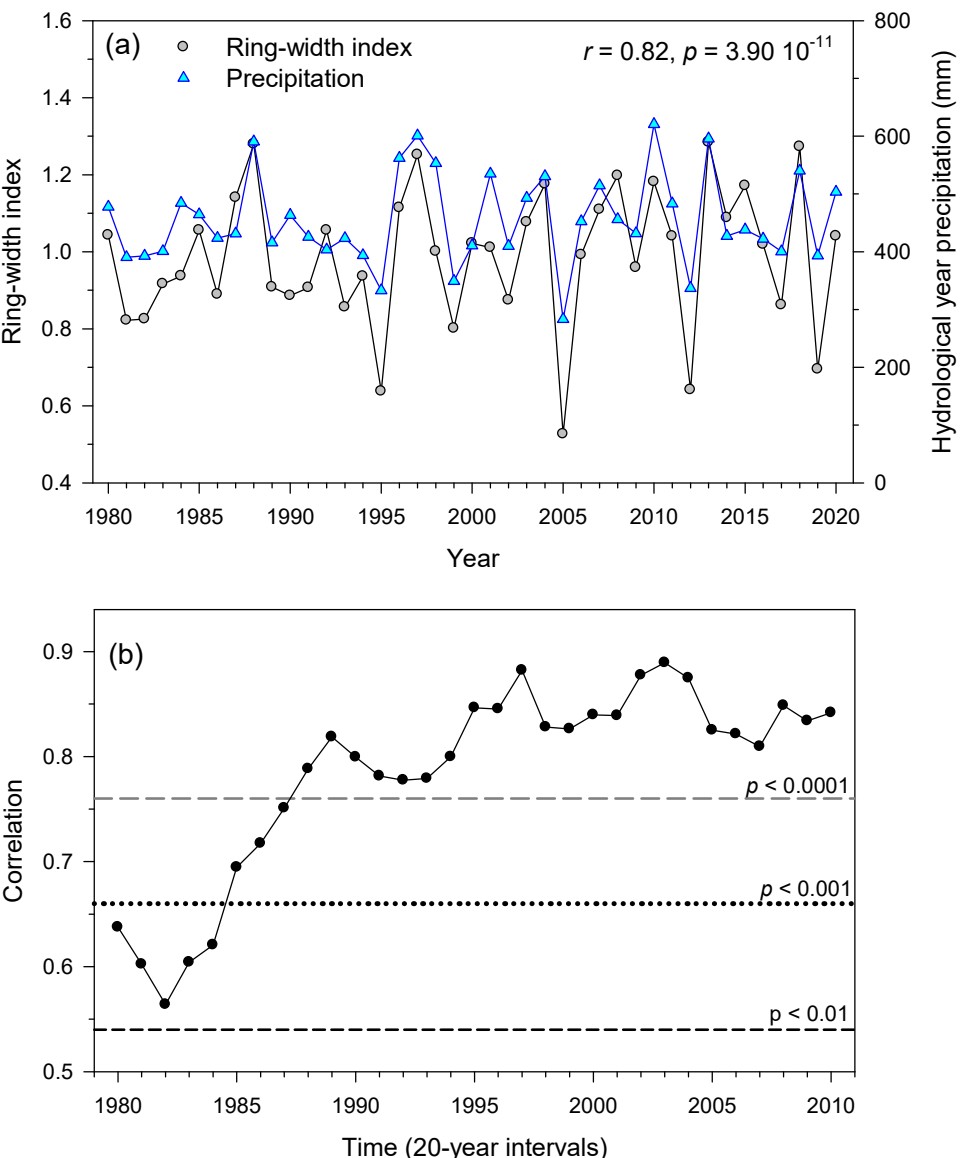

**Figure 5.** Atlas cedar growth is driven by the total precipitation during the hydrological year (**a**) and this association has strengthened through time (**b**). In the panel (**a**), the ring-width index is the mean of all sites' series, and the precipitation was computed for the region delimited by coordinates 5.75° W–1.00° E and 37.00°–44.25° N. In the panel (**b**), moving Pearson correlations between the two variables shown in panel (**a**) are calculated considering 20-year moving intervals from 1961 to 2020. The horizontal lines show the significance levels.

Regarding climate indices, we found negative correlations between the winter NAO and growth in the southern Puerto de la Mora and Baza sites (Figure S5). In Berriozar, growth rates and the summer WeMO were negatively related, but this relationship was observed for prior-winter WeMO values in Sariñena or for April WeMO values in Bañón. High February SOI values were positively related to growth in Sariñena and Bañón, whereas high prior autumn SOI values were negatively related to growth in Monte Mario-Béjar.

The climate–growth correlations calculated on the first three principal components of the PCAs, particularly the PC1, highlighted wet winter and spring conditions and cool summer conditions as the most important to enhance Atlas cedar growth (Figure S6a). In the case of climate indices, high NAO and WeMO values in the prior winter were negatively related to the PC1, whereas high SOI values in February and August were positively related

(Figure S6b). The PC2 showed negative correlations with summer WeMO, but the PC3 showed positive correlations with spring WeMO and summer NAO.

### 3.3. Models of Resilience Indices at Individual Level

The model for resistance (Rt) included tree age and mean sensitivity as fixed effects both with a positive effect (Tables 3, S1 and S2, Figure 6a,b) and accounted for 37% of the variation in Rt ($R^2$m = 0.34; $R^2$c = 0.37). The model of the recovery (Rc) component showed significant positive effects of ring width and, mainly, mean sensitivity (Tables 3 and S1), and accounted for 84% of the variation in the data ($R^2$m = 0.83; $R^2$c = 0.84; Figure 6c,d). Lastly, the resilience (Rs) model only showed a significant positive influence of tree age (Table 3; Figure 6e) and accounted for 32% of the variation ($R^2$m = 0.31; $R^2$c = 0.32). Overall, the variability accounted for random effects was low, given that models were run on standardized ring-width indices.

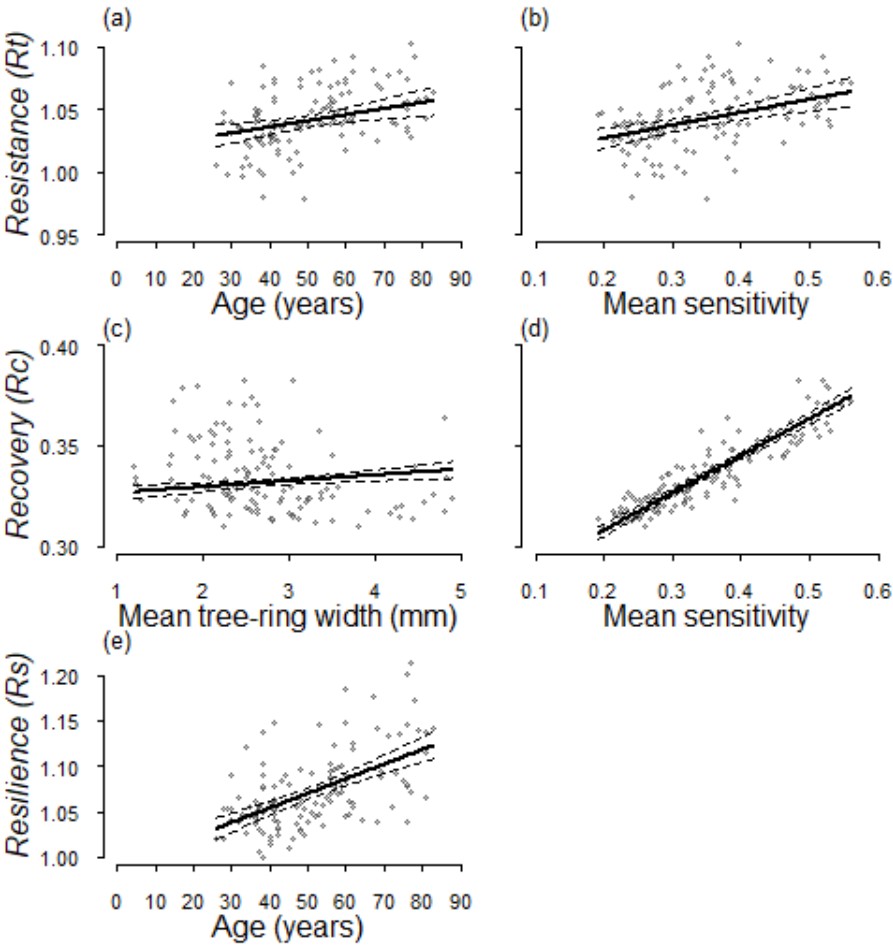

**Figure 6.** Effects of the fixed effects on the resistance (**a**,**b**), recovery (**c**,**d**) and resilience (**e**) components according to the linear mixed-effect models. The grey points indicate the values observed for each individual in the studied populations (*n* = 130 trees) and the solid line indicates the effect of each fixed effect on the resilinece components (dashed lines represent the upper and lower confidence intervals).

**Table 3.** Results of the linear mixed-effect models applied on resilience indices (Rt, resistance; Rc, recovery; Rs, resilience). The table shows the coefficients associated to different fixed factors with their standard errors (SE), *t* statistics of each factor (Age, tree age; MSx, mean sensitivity; TRW, mean tree-ring width) and significance levels (*p*).

|  | Coefficient | SE | df | t | p |
|---|---|---|---|---|---|
| Rt |  |  |  |  |  |
| Intercept | 0.983 | 0.010 | 120 | 101.168 | <0.001 |
| Age | 0.002 | 0.001 | 120 | 2.989 | 0.003 |
| MSx | 0.103 | 0.025 | 120 | 4.135 | <0.001 |
| Rc |  |  |  |  |  |
| Intercept | 0.263 | 0.005 | 120 | 52.287 | <0.001 |
| MSx | 0.186 | 0.009 | 120 | 20.451 | <0.001 |
| TRW | 0.003 | 0.001 | 120 | 3.027 | 0.002 |
| Rs |  |  |  |  |  |
| Intercept | 0.990 | 0.011 | 121 | 86.948 | <0.001 |
| Age | 0.002 | 0.000 | 121 | 7.451 | <0.001 |

## 4. Discussion

Quantifying growth sensitivity to drought and understanding how this may affect the patterns of carbon uptake and climate change mitigation on mountain conifer plantations is relevant under ongoing climate warming. We focused on European Atlas cedar plantations subjected to seasonal drought to quantify their growth resilience and to investigate the effects of local climate conditions and individual features (e.g., tree age) on resilience. Our results support contrasting growth responsiveness to local climate and drought events and the significant influence of tree-level traits on growth resilience. As hypothesized, the plantations situated in the driest sites from south-eastern Spain showed the highest responsiveness to precipitation (Figure 2), including the highest percentage of growth variability explained by the precipitation of the hydrological year (61% in Baza, cf. Table 2), and the highest correlations between growth rates and the SPEI drought index (Figure 4). This confirmed our first hypothesis. The most extreme values of the resilience components were not always observed in these southernmost dry sites since some mid-latitude sites, such as Bañón, showed a high responsiveness to climate and elevated resilience, whereas others, such as Sariñena, showed an unexpected low responsiveness to precipitation despite several droughts having severely impacted its growth (Figures 1, 2 and S4). These discrepancies indicate local factors, such as the soil type or genetic origin of the seeds, modulating growth responses to climate (e.g., [53]). It is probable that the provenance of the seeds have some effect on the growth–climate relationships, as it happens in the species' natural distribution area in Morocco and Algeria [21,22], but, regrettably, we lacked accurate information on the geographical origin of the seeds used in each plantation. Further studies could combine molecular markers with tree-ring data to solve this issue.

At the individual level, the tree age, growth rate and year-to-year growth variability (mean sensitivity) significantly impacted growth resilience, thus confirming our second hypothesis (Table 3). As expected, resistance and resilience increased as tree age did, whilst resistance and recovery increased as growth variability did (Figure 6). As observed in other conifers, abrupt growth reductions are common features of Atlas cedar growth [54].

The geographical gradient observed in the climate–growth relationships (Figure 2) probably reflects the phenological differences in radial growth and xylogenesis controlled by warmer, sunnier and drier conditions southwards, shifting the growth onset towards sooner dates as has been found under Mediterranean continental conditions [55]. The differences in the onset of *Cedrus libani* A. Rich. cambial activity along an altitudinal gradient were driven by different stem and air temperatures [56]. The high growth rates observed in some of these sites, such as in Puerto de la Mora (Table 2), suggest trees

showed high growth rates in spring and/or longer growing seasons with a potential secondary growth peak after the summer drought. The relevant role played by prior-winter precipitation as a driver of Atlas cedar growth in these sites from semi-arid south-eastern Spain (Figure 2) can be explained by the replenishment of soil water pools or by the early start of the dry season leading to the fast growth in early spring. The observed climate–growth correlations agree with previous studies on natural forests of Atlas cedars in North Africa [18,21,22,57], and also with other Mediterranean cedar species such as *C. libani* in Turkey [58].

The northernmost and southernmost sites (Valliguières and Puerto de la Mora) represented the extremes of the climate gradient with the strongest and weakest growth responses to drought, respectively (Figures 3b and 4). However, local outliers were also observed along this gradient. For instance, the wet Berriozar site showed a high year-to-year growth variability (Figures 1 and 3a), which could be explained by past local disturbances, such as logging or pest outbreaks. The mid-latitude Sariñena and Bañón sites showed, respectively, lower and higher growth sensitivity than expected (Figures 2 and 3b). These divergences may be caused by the proximity of the Sariñena stand to irrigated crop lands, which could increase soil moisture levels, as indicated by its high growth rates (Figure S3), and to the extreme continental and dry conditions of the Bañón site which could amplify the trees' responsiveness to drought (Figure 4). Despite the impending dieback symptoms that may be expected in the drier southernmost populations, the local variability observed in the Atlas cedar network supports further research considering additional site information, such as the seed origin, soil depth and texture, disturbances, silvicultural treatments, forest management or competition degree [53]. For example, a higher surface rock cover and soil stoniness increased the soil water holding capacity and the growth resilience and survival in the semi-arid plantations of *Pinus halepensis* from Israel [59]. A recent meta-analysis showed that higher competition strengthened the association between the water availability and year-to-year growth rates, reduced resistance and improved recovery, but did not consistently affect resilience in the same direction, i.e., the relationships between competition and resilience was not always negative [24].

Interestingly, Atlas cedar growth is becoming increasingly dependent on the hydrological-year precipitation at the regional scale (Figure 5), while this strengthening association may be controlled by changes in the atmospheric circulation patterns, as captured by the winter NAO (Figure S6b). The increasingly stronger coupling between regional precipitation and Atlas cedar growth variability may be related to the recurrent droughts occurring in the late 20th and early 21st centuries (Figures 1 and 5). Our analyses indicate that the precipitation–drought coupling was reinforced by high winter NAO values and low precipitation in winter and spring in response to changes in the sea surface temperature over the northern Atlantic Ocean [33,60]. Similar dry winter-spring conditions linked to the NAO have been shown to reduce Atlas cedar growth in Moroccan forests [61]. It is remarkable that the NAO impact on precipitation and growth was observed even in south-eastern Spain, an area which is less influenced by the NAO than western Spain [62]. Further research considering the NAO and other climate variables (e.g., field measurements of soil moisture) and focusing on selected stands could investigate if the rising temperatures are increasing water-use efficiency or the atmospheric water demand (vapor pressure deficit) by using C and O isotopes. This would contribute to better understanding to what extent European Atlas cedar plantations are becoming more vulnerable to hotter droughts and, by extension, other drought-prone mountain conifer plantations. Such knowledge is mandatory if we aim to define reliable adaptive management tools, including thinning treatments [63]. We argue that managing some Atlas cedar plantations towards their conversion into multi-purpose, mixed plantations (cf. [64]) with drought-tolerant tree species (e.g., Atlas cedar and holm oak—*Quercus ilex* L.) could also be an efficient and still little-explored alternative tool for climate adaptation strategies.

## 5. Conclusions

The growth of Atlas cedar plantations from drought-prone areas in south-western Europe was sensitive to water shortage, particularly from the prior winter to the current spring, but with strong variations from site to site. Consequently, the growth rate and resilience of these planted stands were related to their local climatic conditions. For instance, the growth of plantations from the driest sites in south-eastern Spain was strongly coupled to winter precipitation and seems to be constrained by long-lasting droughts. In contrast, the growth of plantations from the wettest sites in south-eastern France and northern Spain were more related to spring and summer precipitation and it appears slightly constrained by shorter drought periods. We found local exceptions to this general, mainly latitudinal, climatic pattern. At the tree level, resilience is linked to tree age and year-to-year growth variability. Regionally, the growth of Atlas cedar is becoming strongly coupled to precipitation and this coupling is associated to changes in the winter NAO. Impending dieback symptoms were observed in the southernmost plantation (Puerto de la Mora), where declining growth trends and limited resilience suggest that current climate stress surpasses the Atlas cedar tolerance threshold. The analyses of growth resilience at the stand and tree levels allow disentangling different drivers of growth changes in planted tree populations and understanding how climate extremes, such as drought, limit radial growth and the capacity of plantations to act as effective carbon sinks.

**Supplementary Materials:** The following are available online at https://www.mdpi.com/article/10.3390/f12121751/s1, Figure S1: Views of (a) two Atlas cedar plantations, and (b) location of the study sites in south-eastern France and Spain, Figure S2: Climate diagrams and water balance plots from selected study sites (see sites' abbreviations in Table 1), Figure S3: Series of basal area increment for the eight study Atlas cedar plantations plotted as a function of year or cambial age (in years), Figure S4: Resilience indices (Rc, recovery; Rt, resistance; Rs, resilience) calculated for the study Atlas cedar plantations, Figure S5: Climate indices-growth relationships assessed in eight Atlas cedar plantations (see sites' codes in Table 1), Figure S6: (a) Climate– and (b) climate indices–growth relationships assessed in the first principal components (a, PC1; b, PC2; and c, PC3) calculated on the matrix of ring-width series from eight Atlas cedar plantations (see sites' codes in Table 1). Table S1: Spearman correlation coefficients calculated at the individual level ($n$ = 130 trees) in Atlas cedar plantations., Table S2: Linear mixed-effect models selection table.

**Author Contributions:** Conceptualization, J.J.C. and A.G.; methodology, J.J.C., M.C. and A.G.; software, J.J.C. and A.G.; validation, J.J.C., A.G., J.C.L. and R.M.N.-C.; formal analysis, J.J.C. and A.G.; data curation, J.J.C. and M.C.; writing—original draft preparation, J.J.C. and all co-authors; writing—review and editing, all authors; funding acquisition, J.J.C., J.C.L., R.M.N.-C., Á.R.-C., F.S., P.-J.D. and F.C. All authors have read and agreed to the published version of the manuscript.

**Funding:** This research was funded by the Spanish Ministry of Economy, Industry and Competitiveness: FORMAL (RTI2018-096884-B-C31) project and SilvAdapt RED2018 102719 T project.

**Acknowledgments:** We acknowledge the E-OBS dataset from the EU-FP6 project UERRA the Copernicus Climate Change Service, as well as the data providers in the ECA&D project (https://www.ecad.eu, accessed 4 October 2021). We acknowledge the institutional support of the Sierra Nevada National Park. We also thank the ERSAF group and, particularly, Antonio Cachinero and Antonio Ariza, for their assistance during this research.

**Conflicts of Interest:** The authors declare no conflict of interest. The funders had no role in the design of the study, in the collection, analyses or interpretation of data, in the writing of the manuscript or in the decision to publish the results.

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
