# Peer review of "Shifting Precipitation Patterns Drive Growth Variability and Drought Resilience of European Atlas Cedar Plantations"

_forests, doi:10.3390/f12121751_

Round 1
Reviewer 1 Report
General comments
The paper is interesting and, as non specialist, I feel the analysis have been done properly and in a rigorous scientific way. The research design is appropriate and it is interesting to have results from populations covering an important portion of the native range of the species.
As modeller and geneticist I believe that the results could have been much more interesting if combined with molecular markers. Actually we are talking about plantations and therefore the provenance of the seeds may have a strong effect on the growth-climate relationship. And the results from artificial stands must be carefully extended to the whole species... However this part is somehow discussed in the Discussion section (L387) but an extension of it would be welcomed. The Authors are not asked to run more analysis from my side at least, just some bibliographic research.
Concerning the climatic data I believe the E-OBS may be a good source but I suggest the Authors to consider (in future) some tailored climatic data that are being delivered from scale-free downscaling systems such as ClimateEU or ClimateDT portal (https://ibbr.cnr.it//climate-dt/). Actually the scale-free downscaling may produce more reliable data than 1km rasters.
However these are just some points I would like to suggest but nothing relevant for the paper itself. Concerning the manuscript I have just few suggestions:
Specific comments
1) I suggest to add a figure showing where the stands you have sampled are located. The table 1 is relevant but something graphic could be more direct and useful for the reader. Then you can consider to add some graphs concerning the climate. Most of this stuff is reported in the supplementary materials already and I encourage the authors to transfer something as figures in the work. I don't want them to cite me but I can suggest a layout such as Figure 1 in this published manuscript: https://www.mdpi.com/1999-4907/10/7/584
2) As non specialist I think the methods are applied properly but I suggest to add some references concerning some tuning such as the 67% cubic smoothing spline and 50% cut-off frequency (L173-174) just to make the reader aware why this tuning and and who else used this before. The same for Pearson correlation: why a linear correlation is expected? Why not a Spearman correlation coefficient?
3) Concerning the results I was expecting from the Authors some information on the Expressed population signal. Even if some works like thi (https://www.sciencedirect.com/science/article/abs/pii/S1125786517300231) questioned its use, I believe it could be a good way to show how variable the samples are within each site and to analyse whether the variance between stands is greater or not than the variance within stands.
4) Last I think that the main findings of your work are not clearly reported. I suggest to reshape the discussion section introducing some basic concept on the genetic structure of Atlas cedar in the Mediterranean basin (from literature of course) but, above all, to start the discussion section with a less general statement. Actually I strongly suggest you to start stating if you accept or reject the research hypothesis.
Hope it helps
Author Response
General comments
The paper is interesting and, as non specialist, I feel the analysis have been done properly and in a rigorous scientific way. The research design is appropriate and it is interesting to have results from populations covering an important portion of the native range of the species.
› We thank you for your positive comments.
As modeller and geneticist I believe that the results could have been much more interesting if combined with molecular markers. Actually we are talking about plantations and therefore the provenance of the seeds may have a strong effect on the growth-climate relationship. And the results from artificial stands must be carefully extended to the whole species... However this part is somehow discussed in the Discussion section (L387) but an extension of it would be welcomed. The Authors are not asked to run more analysis from my side at least, just some bibliographic research.
› We agree with your point but, regrettably, we lack genetic information and historical data on the origin of the seeds used in each plantation. Probably, most used seeds were from Moroccan stands (albeit, we cannot confirm this), but the Atlas cedar is widely distributed across Morocco (Rif, Middel Atlas and High Atlas). As you suggested we add a comment on this issue.
Concerning the climatic data I believe the E-OBS may be a good source but I suggest the Authors to consider (in future) some tailored climatic data that are being delivered from scale-free downscaling systems such as ClimateEU or ClimateDT portal (https://ibbr.cnr.it//climate-dt/). Actually the scale-free downscaling may produce more reliable data than 1km rasters.
› We thank your for your useful comment. We will consider these data in future studies.
However these are just some points I would like to suggest but nothing relevant for the paper itself. Concerning the manuscript I have just few suggestions:
Specific comments
1) I suggest to add a figure showing where the stands you have sampled are located. The table 1 is relevant but something graphic could be more direct and useful for the reader. Then you can consider to add some graphs concerning the climate. Most of this stuff is reported in the supplementary materials already and I encourage the authors to transfer something as figures in the work. I don't want them to cite me but I can suggest a layout such as Figure 1 in this published manuscript: https://www.mdpi.com/1999-4907/10/7/584
› We understand your point. However, we consider that there are enough figures (6) in the main ms. and the table 1 provides a useful summary of climate conditions. Note that we work on plantations so making a map would be less useful than when dealing with natural forests (which, for this species, are mainly located in Morocco and Algeria).
2) As non specialist I think the methods are applied properly but I suggest to add some references concerning some tuning such as the 67% cubic smoothing spline and 50% cut-off frequency (L173-174) just to make the reader aware why this tuning and and who else used this before. The same for Pearson correlation: why a linear correlation is expected? Why not a Spearman correlation coefficient?
› We added comments on these methods, which are widely used in tree-ring sciences. The length of the spline is widely employed in similar studies because it allows removing long- to mid-term frequency variability in growth. Detrended ring-width indices follow a normal distribution which justifies using Pearson correlations.
3) Concerning the results I was expecting from the Authors some information on the Expressed population signal. Even if some works like thi (https://www.sciencedirect.com/science/article/abs/pii/S1125786517300231) questioned its use, I believe it could be a good way to show how variable the samples are within each site and to analyse whether the variance between stands is greater or not than the variance within stands.
› We added information on a similar statistic (correlation with the mean site series or master series), which is widely used in tree-ring sciences to assess the coherence and replication of a mean series or chronology. As you pointed out the use of the EPS is questioned so we opted by using and discussing the mean correlation with the master or site series.
4) Last I think that the main findings of your work are not clearly reported. I suggest to reshape the discussion section introducing some basic concept on the genetic structure of Atlas cedar in the Mediterranean basin (from literature of course) but, above all, to start the discussion section with a less general statement. Actually I strongly suggest you to start stating if you accept or reject the research hypothesis.
Hope it helps
› We started the discussion by accepting the hypothesis. We understand your interest on genetics but note that other factors (local site conditions such as soil type, tree-to-tree competition) could be also as important (or more important) modulators of climate-growth relationships. As mentioned before, we lack information on these sources of variability so we prefer not to focus on the genetic structure of the species and rather comment this factor which may be as important as the other modulators.
›Thanks for your comments.
Reviewer 2 Report
Review of the submitted manuscript entitled Shifting precipitation patterns drive growth variability and drought resilience of European Atlas cedar plantations
The submitted manuscript presents the results of a dendrochronological study conducted on two Atlas cedar plantations in southern Europe. Considering observed climate change, this north African species with a small geographic range can be a valuable alternative to native species, given its drought resistance.
The introduction presents and justifies the conducted research. Goals and hypotheses are rational. The description of the research subject is sufficiently detailed. I have no objection to the materials and methods. I have only a small comment on the presentation of the results, wherein the description of figure 3 is information about what the biplots show. However, they are not present in this figure. Nice Discussion! The conclusions are rational and follow from the obtained results.
Kind regards
Author Response
The submitted manuscript presents the results of a dendrochronological study conducted on two Atlas cedar plantations in southern Europe. Considering observed climate change, this north African species with a small geographic range can be a valuable alternative to native species, given its drought resistance.
The introduction presents and justifies the conducted research. Goals and hypotheses are rational. The description of the research subject is sufficiently detailed. I have no objection to the materials and methods. I have only a small comment on the presentation of the results, wherein the description of figure 3 is information about what the biplots show. However, they are not present in this figure. Nice Discussion! The conclusions are rational and follow from the obtained results. Kind regards
› We added information on the biplots presented in the legend of Figure 3 to improve its presentation. Thanks a lot for your positive comments on our study.